# Evaluation of IoT-Enabled Monitoring and Electronic Nose Spoilage Detection for Salmon Freshness During Cold Storage

**DOI:** 10.3390/foods9111579

**Published:** 2020-10-30

**Authors:** Huanhuan Feng, Mengjie Zhang, Pengfei Liu, Yiliu Liu, Xiaoshuan Zhang

**Affiliations:** 1College of Engineering, China Agricultural University, Beijing 100083, China; fenghh@cau.edu.cn (H.F.); zhmystic@cau.edu.cn (M.Z.); S20193071146@cau.edu.cn (P.L.); 2Beijing Laboratory of Food Quality and Safety, China Agricultural University, Beijing 100083, China; 3Department of Mechanical and Industrial Engineering, Norwegian University of Science and Technology, 7491 Trondheim, Norway; yiliu.liu@ntnu.no

**Keywords:** IoT-enabled monitoring system (IoTMS), electronic nose, salmon, Convolutional Neural Networks and Support Vector Machine (CNN-SVM), freshness

## Abstract

Salmon is a highly perishable food due to temperature, pH, odor, and texture changes during cold storage. Intelligent monitoring and spoilage rapid detection are effective approaches to improve freshness. The aim of this work was an evaluation of IoT-enabled monitoring system (IoTMS) and electronic nose spoilage detection for quality parameters changes and freshness under cold storage conditions. The salmon samples were analyzed and divided into three groups in an incubator set at 0 °C, 4 °C, and 6 °C. The quality parameters, i.e., texture, color, sensory, and pH changes, were measured and evaluated at different temperatures after 0, 3, 6, 9, 12, and 14 days of cold storage. The principal component analysis (PCA) algorithm can be used to cluster electronic nose information. Furthermore, a Convolutional Neural Networks and Support Vector Machine (CNN-SVM) based algorithm is used to cluster the freshness level of salmon samples stored in a specific storage condition. In the tested samples, the results show that the training dataset of freshness is about 95.6%, and the accuracy rate of the test dataset is 93.8%. For the training dataset of corruption, the accuracy rate is about 91.4%, and the accuracy rate of the test dataset is 90.5%. The overall accuracy rate is more than 90%. This work could help to reduce quality loss during salmon cold storage.

## 1. Introduction

Salmon is a popular and healthy food choices that contains rich minerals and unsaturated fatty acids. However, salmon is a highly perishable food and has a short life [1,2], which means it deteriorates easily, and the process is accelerated along with the environmental change due to numerous factors such as the processing method, package material, cold ambient condition, and preservation technology [3,4]. Freshness level is an important reference for market value. Therefore, cold storage using intelligent monitoring technology to keep it at a low-temperature condition contributes to improving salmon quality value. It is necessary to develop a strategy to monitor and detect spoilage of salmon during cold storage.

Salmon freshness varies with time and environment. Thus, an important aspect of fish products transportation and consumption is the effective monitoring of time and environment conditions, which affect overall quality of fish [5,6]. The chemical indicators: pH, odor and, sensory evaluation, show that the product quality has a high correlation degree with temperature changes. As a consequence, temperature has an important effect on salmon meat, and monitoring and traceability is especially important in its cold storage condition [7]. In order to collect the real-time ambient information and enhance salmon quality and safety, Internet of thing (IoT) based monitoring and spoilage detection are becoming the focus of the current research demand [8,9].

With the development of information technology, sensors devices and electronic noses are used for food quality monitoring and detection. Sensors are an interdisciplinary, integrated, cutting-edge field of research that integrates embedded computing, network and wireless communications technologies, and distributed processing [10,11,12]. It can realize the requirements of information transmission, processing, display, and control with low cost for the monitoring objects. An electronic nose (e-nose) is a rapid and feasible detection device equipped with sensors array, which are powerful tools in food quality assessment. It can quickly detect the odor with high sensitivity in complex samples. However, the electronic nose still has some technical problems, such as high-power consumption and large size. The advantages and limitations of e-nose application on food assessment are listed in Table 1. Electronic nose freshness detection will be waved in the process of nitrogen, alcohol, amine, ammonia, and sulfur compound gases, as these gas concentrations and meat freshness are associated with the gas sensor array of these volatile odor response characteristics [13,14]. It can achieve freshness identification.

After data acquisition, effective and high-precision processing and fusion are very important. Deep learning-based algorithms enable qualitative data [22]. Currently, food quality freshness evaluation methods can be further processed by establishing classification and regression algorithms, such as k-nearest neighbors (KNN), Principal component analysis (PCA), fuzzy c-means, artificial neural network (ANN), Partial least squares (PLS), and Convolution neural network (CNN) [23,24]. The algorithms are used to realize quantitative analysis for quality parameters prediction, shelf-life prediction, and freshness grading [25,26].

Based on the actual stage of research, a successful cold storage needs an automated and efficient monitoring and spoilage detection system in order to assist in managing the collected information. In this study, the objective is to evaluate IoT-enabled monitoring and electronic nose spoilage detection for quality parameters changes and freshness under cold storage conditions. This leads us to address the following research questions:(1)How can IoT-enabled monitoring and electronic nose spoilage detection be designed and implemented for high precision data capturing?(2)How can we choose quality parameters to analyze quality at different temperatures?(3)How can we build a freshness grading model for salmon samples?

Therefore, the approach of applying monitoring based on sensor and electronic nose is used to capture quality information, and quality parameters, i.e., texture, color, sensory, and pH changes are measured and evaluated. After that, a CNN-SVM based algorithm is used to cluster the freshness level of salmon samples stored in a specific storage condition. Salmon is taken as the experiment subject for the objective to test and evaluate quality storage at 0 °C, 4 °C, and 6 °C. This work could contribute to improve food quality management.

## 2. Materials and Methods

### 2.1. Quality Monitoring and Detection System Design

#### 2.1.1. The Architecture Design of IoT-Enabled Monitoring System (IoTMS)

The proposed IoTMS is used to monitor the cold storage ambient information including temperature value, humidity value, and gas value. The IoTMS is composed of three layers: the data sensing layer, data transmission layer, and application layer. Figure 1 illustrates the IoTMS architecture.

The data sensing layer is mainly to collect ambient information. The sensing sensor devices (Tem & RH sensor and gas sensors mainly including O_2_, CO_2_, and H_2_S sensors) are deployed at the cold storage. The ranges of the temperature, relative humidity, and H_2_S are from −40 to 80 °C, 0 to 100%, and 0 to 100 ppm, respectively. In order to reduce or eliminate errors in the measurement process of the gas sensor, the sensor needs to be tested and calibrated. The sensors calibration curve is shown in Figure 2. O_2_, CO_2_ and SO_2_ measured the output voltage under the condition of different concentration ratios, and the output voltage of each gas sensor showed a good linear relationship with the gas with different volume fractions, and the correlation coefficient R^2^ was above 0.99.

The data transmission layer provides complete and reliable data access services. In IoTMS, the data transmission layer is composed of the database server, application server, the routers, and the firewalls. The database server communicates with the gateway of the remote layer via Local Area Network (LAN). All data, including environmental information and location information, are uploaded from the gateway. All data are used as input to the business logic in the IoTMS, as well as feedback from the automatic control systems of the cold storage and the Internet [27].

The application layer is mainly used for providing the visualization display to easily manage and use data for end-users via personal computer (PC), Smart phone, tablet, etc. In this layer, the captured sensor information can be provided to users in real-time. The operation and configuration interface is easy and friendly to use for industry managers.

#### 2.1.2. The Architecture Design of Electronic Nose Detection System (ENDS)

Electronic nose detection system (ENDS) is composed of gas sensor array, closed detection gas chamber, data acquisition and processing module, and display interface. The block diagram of electronic nose detection system is shown in Figure 3. The sensor array consists of MQ136, MQ 137, MQ 138, TGS2612, TGS822, and TGS2600 sensors. The sensitivity is 1~200 mL/m^−3^, 5~500 mL/m^−3^, 1~25%LEL, 50~5000 mL/m^−3^, 0~100 mL*/*m^−3^, respectively. The operating voltage of the sensor is 5 V. The core controller is the STC12C5A60S2 chip. The signal is converted into a digital signal and sent to the upper computer through a serial port [8]. The upper computer performs feature extraction and pattern recognition of the captured digital signal, and outputs the identification results in the display interface.

### 2.2. Experimental Samples

Salmon samples was directly purchased from the processing company in Shandong Oriental Ocean Sci-tech co., Ltd., Yantai, China). The salmon were transported on ice to the laboratory. A total of 126 salmon samples were used for this work. Then each salmon sample was cut into 30 mm × 30 mm pieces and was vacuum packaged using an low density polyethylene (LDPE) plastic bag. The samples were divided into three groups in an incubator set at 0 °C, 4 °C, and 6 °C respectively. The salmon samples were analyzed after 0, 3, 6, 9, 12, and 14 days of cold storage. The experiment was randomly selected from three groups under different conditions to reduce the error of the individual samples.

### 2.3. Quality Parameters Analysis

#### 2.3.1. Texture Analysis

Texture profile analysis (TPA) of salmon was performed by using a texture analyzer (Model T TA-XT PLUS, Texture Analyzer, Stable Micro System, Godalming, UK) according to the method illustrated by [28,29]. Test speed range is 0.01–40 mm/s. Test distance accuracy is 0.001 mm. Test force accuracy is 0.1 g. TPA includes hardness, chewiness, springiness, cohesiveness, and gumminess. Each sample is measured at 5 points, and the average is used for measurement determination.

#### 2.3.2. Color Properties

The measurement of color difference value was performed by using a colorimeter (Konica Minolta CR-400, Tokyo, Japan), with the settings at 8 mm aperture, 2° observer, D65 light source, and pulsed xenon lamp as default light sources to test salmon samples. The L* (lightness), a* (red to green), and b* (yellow to blue) tristimulus color values were measured according to the CIE (Commission Internationale d’Eclairage of France) color system, calibrated daily with a white calibration plate before use (L* = 97.57, a* = −1.08 and b* = 1.25). In order to measure color of salmon samples, three readings per sample were randomly chosen for measurement every day at the same time. The average values of salmon measurement were used for changes analysis [30,31].

#### 2.3.3. pH Measurement

The pH was measured by using a hand-held pH meter (Detto testo 205, testo AG, chwarzwald, Germany). The pH measurement range was 0 ~ 14 pH. The measurement accuracy range was 0–0.02 pH. Five points of each salmon sample were selected for parallel detection. Three readings per sample were randomly chosen for measurement every day at the same time, and the average value was taken as the result of pH determination.

#### 2.3.4. Sensory Analysis

Sensory analysis was carried out in 5 groups by 25 staff members with untrained panelists who volunteered in the sensory evaluation of the meat. Salmon samples were stored at different temperatures (0 °C, 4 °C, and 6 °C). The test samples were evaluated at room temperature 23 °C under white fluorescent light. To control the identification bias, each raw salmon sample was labeled with a random selection and provided to members of the panelists. The flavor, color, tenderness, and overall acceptance were evaluated by panelists using a ten-point hedonic scale, where 10 points was the highest quality level, and 1 point was the lowest quality level.

### 2.4. Statistical Analysis

Duplicate measurements were executed three times for each experiment at different temperatures (0 °C, 4 °C, and 6 °C) for all parameters. The Pearson correlation coefficient was used to determine the relationship between parameters. The electronic nose data were evaluated by Principal Component Analysis (PCA). It can reduce the dimension of the original data and simplify the analysis of the original data without loss of original data. The parameter information was analyzed and evaluated by the analysis of variance (ANOVA) processing using Origin program (Origin Pro 9.0 software, Origin Lab Corporation, Massachusetts, MA, USA). The differences at *p*-values (*p* < 0.05) were considered significant.

### 2.5. Deep Learning-Based Freshness Grading and Evaluation

The freshness of salmon is affected by many quality parameters including temperature, temperature, pH, order, and texture changes during cold storage. Meat spoilage is an irreversible process as its freshness level gradually decreases with the extension of storage time and cannot change back. In this work, the salmon freshness was divided into three grades: fresh, acceptable, and corruption.

To effectively identify salmon freshness, we adopted the improved the convolutional neural network and support vector machine (CNN-SVM) algorithm to achieve freshness grading and evaluation of salmon during cold storage. CNN is a feedforward neural network, with each convolution layer consisting of a filter layer, a nonlinear layer, and a spatial sampling layer. CNN has a higher accurate extraction of data information. SVM is more effective for solving classification prediction problems, and it has a good effect in learning complex nonlinear equations [32,33]. The specific steps of CNN-SVM modeling are as follows:Step 1: Collect the salmon data samples;Step 2: Divide the data set into training samples and test samples;Step 3: Establish CNN structure;Step 4: Training CNN model with training samples;Step 5: Feature extraction and SVM model training;Step 6: Replace the trained SVM model with the softmax layer;Step 7: Use test samples to evaluate the performance of the model CNN-SVM.

## 3. Results and Discussion

### 3.1. Monitoring Data Changes Analysis

#### 3.1.1. Ambient Data Dynamic Changes Analysis

Salmon meat have strict range of temperature and humidity requirements during cold storage, which has a great influence on monitoring and detecting quality changes. A real experiment was studied in a 14-day in incubator temperature set at 0 °C, 4 °C, and 6 °C. The temperature and humidity data dynamic changes in a 14-day by using sensor monitoring in real-time is shown in Figure 4.

Ambient temperature changes ranged from about −1.9 °C to 1.6 °C storage at 0 °C. Humidity changes were small, and ranged from about 26.2–34.3%. When the incubator was set at 4 °C, the temperature changes ranged from about 2.8 °C to 5.9 °C, and the humidity ranged from about 27.4–32.6%. When the incubator was set at 6 °C, the temperature sensor data changes ranged from about 4.8 °C to 7.7 °C, and the humidity ranged from about 25.4% to 31.7%. The temperature distribution showed that the amplitude of the temperature value was less than 2 °C. Therefore, temperature and humidity changes were relatively stable and reliable to provide a safer environment for experimental accuracy.

#### 3.1.2. Electronic Nose Data Analysis by PCA Method

PCA was used to identify electronic nose dataset at 0 °C, 4 °C, and 6 °C. It contributes to reducing the dimensionality and the redundancy of the sensor dataset [34,35,36]. According to the score of principal components, the first two main components (which will contain most of the information) can be represented in a plot. The PCA analysis results are shown in Figure 5.

Figure 5a–c represents the number of storage days at 0 °C, 4 °C, and 6 °C on a two-dimensional plot. The first two PCs can be used to represent 99.4% from the electronic nose data variance (Figure 5a) of storage days at 0 °C, 97.8% from the electronic nose data variance (Figure 5b) of storage days at 4 °C, and 98.3% from the electronic nose data variance (Figure 5c) of storage days at 6 °C. There was little variance in odor information of salmon samples at 0 °C. There are an overlaps among day 6 and day 5, and day 8 and day 9 stored at 4 °C, and there is also an overlap between 0 °C and 4 °C. Therefore, it was found that PCA can distinguish different characteristic odors.

### 3.2. Quality Parameters Change Analysis

#### 3.2.1. Texture Change

The texture properties of salmon changed over time and temperature. Six metrics are analyzed using texture profile analysis (TPA) including hardness, chewiness, springiness, cohesiveness, gumminess, and resilience. Figure 6 shows the results of TPA properties of salmon samples stored at refrigerated temperature (0 °C, 4 °C, and 6 °C) after 0, 3, 6, 9, 12, and 14 days of cold storage.

As shown in Figure 6, the overall trend of salmon TPA change decreased with increase of storage time. This may be due to the fact that after the death of salmon, the muscles become stiff and then dissolve, eventually leading to a decrease in TPA. There was a significant change (*p* < 0.05) in the TPA of salmon meat stored at 6 °C for 14 days, and R^2^ > 0.90. There was significant change (*p* < 0.05) in the hardness parameter of salmon meat stored at 4 °C during the first 9 days of storage time. However, there was no significant change in the hardness parameter of salmon meat stored at 0 °C during the first 6 days of storage time. There was significant change (*p* < 0.05) in the chewiness parameter of salmon meat stored at 4 °C after 3 days of storage time and stored at 0 °C and 6 °C for 14 days. There was significant change (*p* < 0.05) in the springiness, cohesiveness and gumminess parameter of salmon meat stored at 0 °C and 4 °C after 3 days of storage time. There was significant change (*p* < 0.05) in the resilience parameter of salmon meat stored at 0 °C and 4 °C after 3 days of storage time. Therefore, TPA changes are highly correlated with time and temperature (R^2^). The temperature has important effected on texture properties.

#### 3.2.2. Color Change

The color of salmon meat will change under different temperature condition. When salmon spoilage is higher, the color change is more significant. The color properties of L*, a*, and b* valued change results are illustrated in Figure 7.

Color is an important visual assessment characteristic for food. The overall trend L* and b* value change decreased during storage time. There was a significant change (*p* < 0.05) in the L* of salmon samples stored at 4 °C for 14 days. However, there was no significant change at 0 °C during the first 6 days of storage time, and there was no significant change at 6 °C on 6 day and 9 day of storage time. The overall trend a* value change increased during storage time. There was a significant change (*p* < 0.05) in the a* of salmon samples stored at 0 °C and 6 °C for 14 days. There was no significant change at 4 °C during the after 9 days of storage time. There was a significant change (*p* < 0.05) in the b* of salmon samples stored at 4 °C for 14 days. There was no significant change at 0 °C on the 9 day and 12 day of storage time. There was no significant change at 6 °C on the 6 day and 9 day of storage time. Therefore, the temperature has an important effect on color characteristics, and color changes are correlated with time and temperature (R^2^ > 0.75).

#### 3.2.3. pH Change

Changes in pH value of salmon are shown in Figure 8. pH values at different temperature showed the same changes trend in which it decreased in the first 3 days and then increased after 3 days. It was close to the results of [37]. This may be due to lactic acid produced by glycolysis or carbon dioxide absorption by air or microorganism. Meanwhile, with the increase of time, fish will produce an enzymatic reaction, and this leads to the decrease of pH value. The change of temperature has an important effect on the pH value of salmon. There was a significant change (*p* < 0.05) in the pH parameter of salmon samples stored at 0 °C, 4 °C, and 6 °C for 14 days.

#### 3.2.4. Sensory Change

Sensory change analysis of the flavor, color, tenderness, and overall acceptance at different storage temperatures (0 °C, 4 °C, and 6 °C) is shown in Figure 9. Sensory properties of all salmon samples were satisfactory on the first 3 days of the storage time. Significant differences in overall acceptance of salmon samples (*p* < 0.05) were observed during cold storage. The sensory scores show a downward trend with time and temperature. The sensory score decreased with the cold storage time, and the sensory score decreased also with the temperature change. It indicates that the higher the temperature, the worse the sensory acceptance.

### 3.3. Freshness Grading and Evaluation Analysis

The freshness grading and evaluation of salmon samples was performed by using CNN-SVM method. The color, texture, pH, and sensory parameters were used as an input dataset of evaluation model. We used 80% training dataset for the training model. Twenty percent of test data were used for model grading and evaluation. After full training, the freshness grading and evaluation was carried out in the test set. The freshness grading results is shown in Figure 10. The freshness evaluation accuracy is demonstrated in Figure 11.

The model realized the freshness grading of salmon verification dataset and test dataset samples at 0 °C, 4 °C, and 6 °C. The CNN-SVM model has high classification accuracy rate for freshness grade than SVM model, which has poor classification accuracy rate for corruption grade. The maximum iteration times is 1026. For the training dataset of freshness, the accuracy rate is about 95.6%, the accuracy rate of test dataset is 93.8%. For the training dataset of corruption, the accuracy rate is about 91.4%, the accuracy rate of test dataset is 90.5%. The overall accuracy rate is more than 90%, which indicates that the model has a good classification results.

### 3.4. System Evaluation

In this paper, salmon quality was combined with multi-sensor sensing and electronic nose-based spoilage detection system designed to survey the performance before and after implementation and provide the basis for quality improvement during cold storage. IoT-enabled monitoring system (IoTMS) could realize the micro-ambient monitoring in real time during cold storage.

The aim of system evaluation is to improve the performance on function requirement, system performance, and energy consumption. Managers and workers have been invited to test and demonstrate how the system should be optimized to high-quality sensors data and control for improving the salmon quality. Through expert evaluation, the system can realize monitoring and transmitting in real time and deploy in a simple and easy way, which has higher efficiency than a wireless sensing system. The electronic nose detection system was feasible for identification ability. However, the disadvantage is to improve sensor data quality and detection accuracy of system.

## 4. Conclusions

This paper presented IoT-enabled monitoring system (IoTMS) and electronic nose detection system. A sensor-based monitoring system was aimed to gather actual ambient temperature and humidity information during cold storage. Electronic nose data were used for detecting freshness of salmon. Principal component analysis (PCA) was used to cluster electronic nose measurements, and quality parameters, i.e., texture, color, sensory, and pH, were measured and evaluated. The CNN-SVM based algorithm is used to cluster the freshness level of salmon samples stored in a specific storage condition. The performances of model and system were comprehensively analyzed and evaluated.

The IoTMS could monitor well the cold chain ambient parameters. PCA can distinguish different characteristic odors from electronic nose signal. The model of CNN-SVM based algorithm has a high classification accuracy rate for freshness grade. The accuracy rate of training dataset of freshness is about 95.6%, and the overall accuracy rate is more than 90%. Therefore, implementation of quality sensing and a spoilage detection system contributes to controlling temperature and reducing quality loss during salmon cold storage.

## Figures and Tables

**Figure 1 foods-09-01579-f001:**
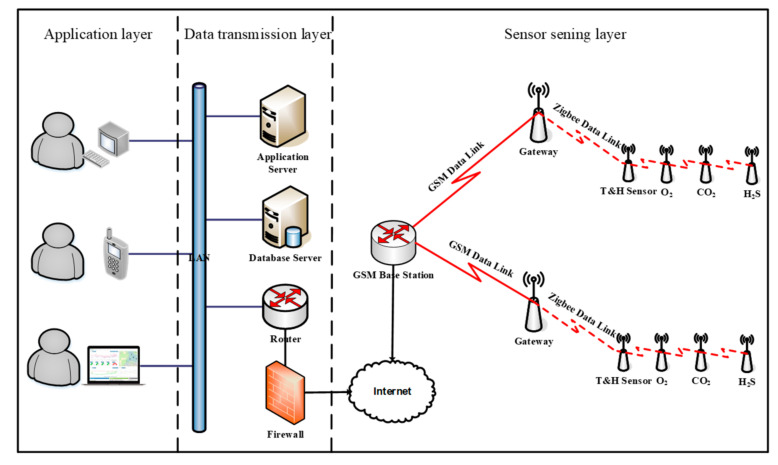
The architecture of IoT-enabled monitoring system (IoTMS). Note: Local Area Network (LAN), Temperature &Relative humidity (Tem&RH).

**Figure 2 foods-09-01579-f002:**
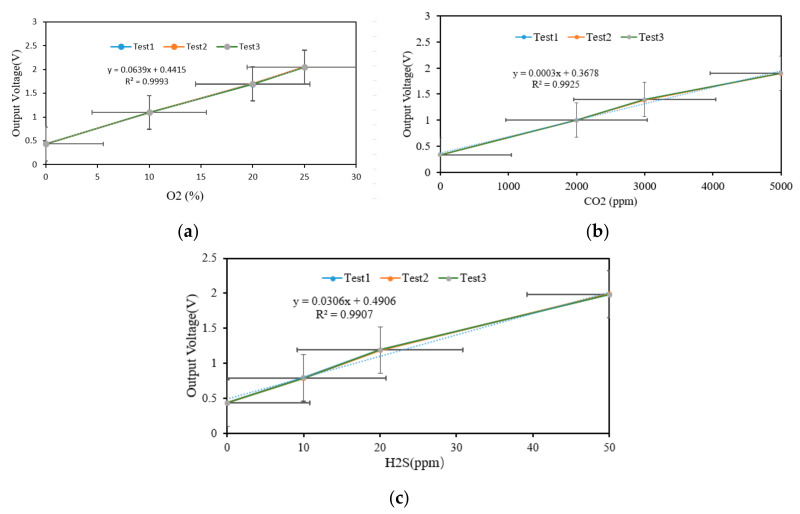
The sensors calibration curves: (**a**) O_2_ sensors calibration curve; (**b**) CO_2_ sensors calibration curve; (**c**) H_2_S sensors calibration curve. Note: x represents gas variables, y represents fitting equation, R^2^ represents the correlation coefficient.

**Figure 3 foods-09-01579-f003:**
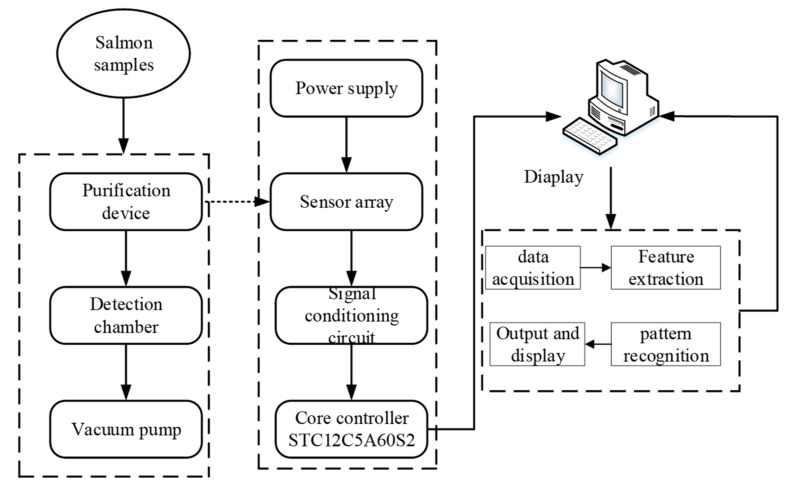
Block diagram of electronic nose detection system.

**Figure 4 foods-09-01579-f004:**
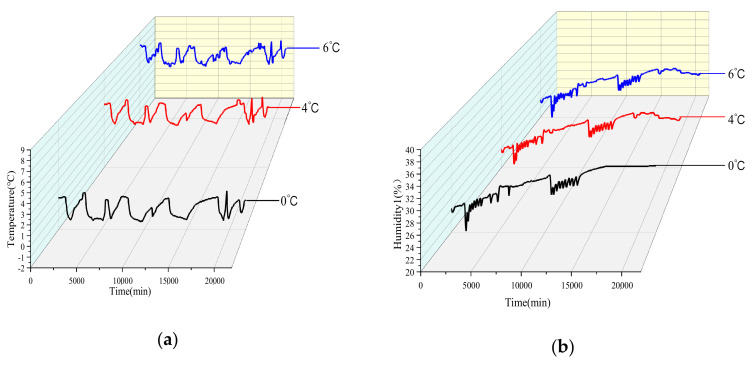
Temperature and humidity dynamic changes of salmon samples stored at refrigerated temperature (0 °C, 4 °C, 6 °C) for 14 days. (**a**) Ambient temperature change; (**b**) Ambient humidity change.

**Figure 5 foods-09-01579-f005:**
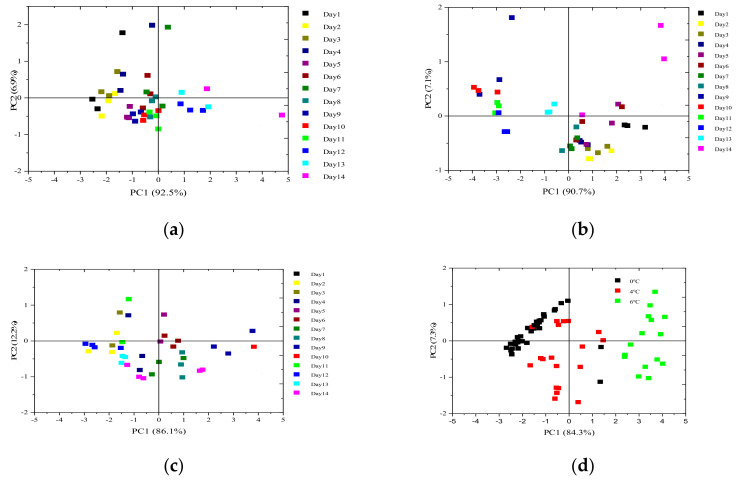
Principal component analysis (PCA) analysis results of the number of storage days. (**a**) PCA scores plot of salmon samples. at 0 °C; (**b**) PCA scores plot of salmon samples at 4 °C; (**c**) PCA scores plot of salmon samples at 6 °C; (**d**) PCA scores plot of the total salmon samples at 0 °C, 4 °C, and 6 °C.

**Figure 6 foods-09-01579-f006:**
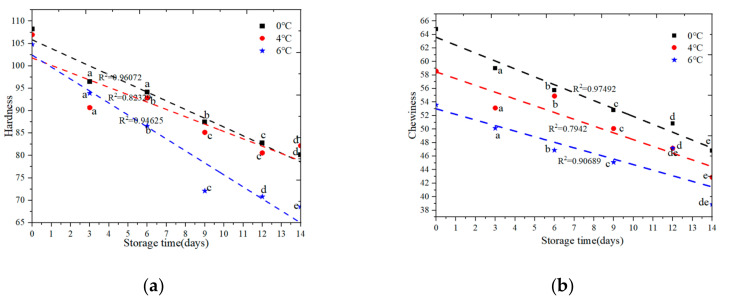
Changes in texture attributes of salmon samples stored at 0 °C, 4 °C and 6 °C, respectively. (**a**) Change in hardness of salmon; (**b**) Change in chewiness of salmon; (**c**) Change in springiness of salmon; (**d**) Change in cohesiveness of salmon; (**e**) Change in gumminess of salmon; (**f**) Change in resilience of salmon. Note: different letters present significant (*p* < 0.05) differences.

**Figure 7 foods-09-01579-f007:**
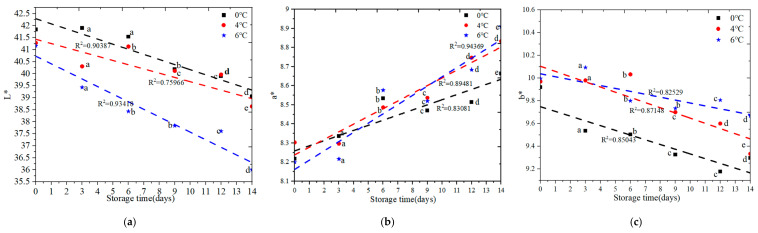
Color change analysis of salmon samples stored at 0 °C, 4 °C, and 6 °C for 14 days respectively. (**a**) L* change analysis; (**b**) a* change analysis; (**c**) b* change analysis Note: different letters present significant (*p* < 0.05) differences.

**Figure 8 foods-09-01579-f008:**
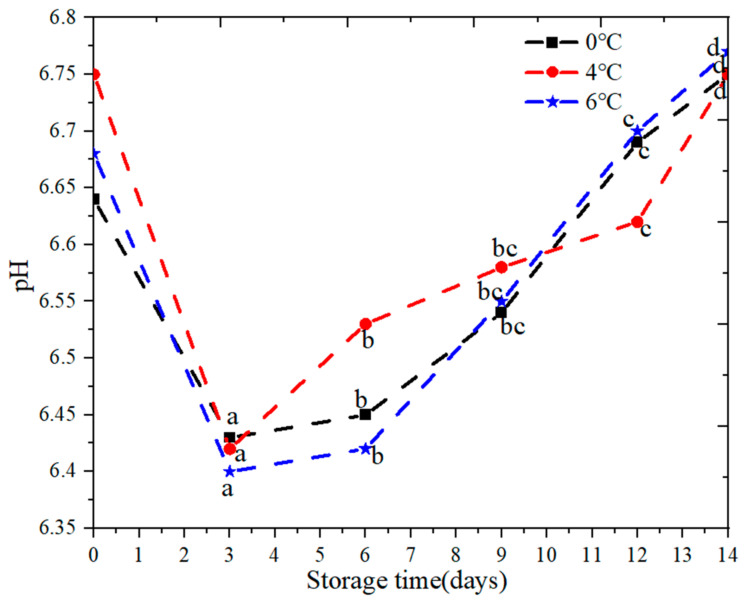
pH change of salmon samples stored at refrigerated temperature (0 °C, 4 °C, 6 °C) for 14 days. Note: different letters indicate significant (*p* < 0.05) differences.

**Figure 9 foods-09-01579-f009:**
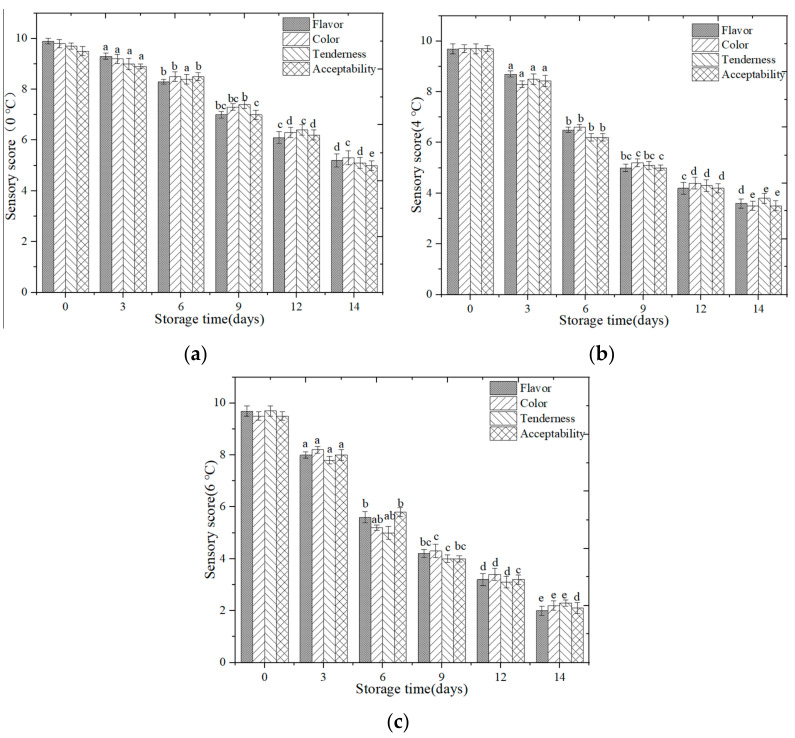
Sensory change of salmon samples (**a**) Sensory change analysis of the flavor, color, tenderness, and overall acceptance at 0 °C; (**b**) Sensory change analysis of the flavor, color, tenderness, and overall acceptance at 4 °C; (**c**) Sensory change analysis of the flavor, color, tenderness, and overall acceptance at 6 °C. Note: Different letters indicate significant (*p* < 0.05) differences.

**Figure 10 foods-09-01579-f010:**
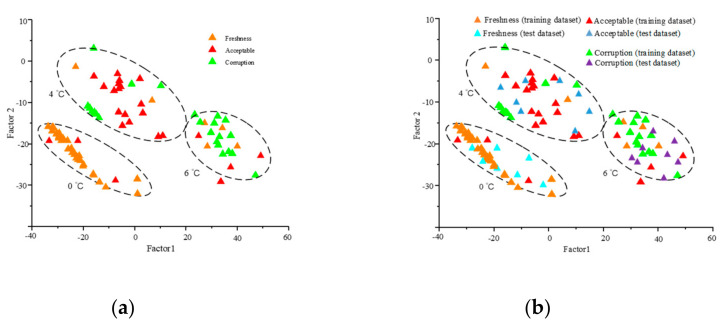
The freshness grading results stored at refrigerated temperature (0 °C, 4 °C, 6 °C). (**a**) The freshness grading results of training dataset; (**b**) the freshness grading results of training dataset and test dataset.

**Figure 11 foods-09-01579-f011:**
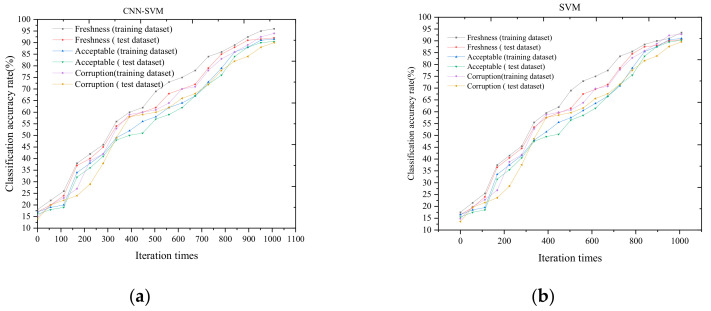
The freshness classification accuracy. (**a**) Convolutional Neural Networks and Support Vector Machine (CNN-SVM) based model classification accuracy rate; (**b**) Support Vector Machine (SVM)based model classification accuracy rate.

**Table 1 foods-09-01579-t001:** The advantages and limitations of e-noses application on food assessment.

Samples	Objectives	Technology Application	Advantages	Limitations	Reference
Meat	Quality Spoilage detecting	Quartz crystal microbalance (QCM)	High sensitivity	High Cost	[15]
Flavor and aroma, Meat flavor	Identification and classification	Neotronics Olfactory Sensing Equipment and QCM	Short response time Low cost	Sensor drift	[16]
Beef	Freshness identification	Electronic nose only	Quick response High sensitivity	High power consumption	[17]
Palm oil	Mixture identification	Electrochemical sensors system	Universal application High sensitivity	Large size High power consumption	[18]
Goat milk	Adulteration Detection	Semiconductor sensors (MOS)	Low cost Low power consumption	Short life span	[19]
Red meat	Spoilage classification	Electronic nose and E-tongue	Quick response Low cost	High power consumption	[20]
Litchi	Freshness evaluation	Electronic nose (E-nose) only	Rapid and non-destructive	Sensor drift High power consumption	[21]

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
