# Peer review of "Evaluation of IoT-Enabled Monitoring and Electronic Nose Spoilage Detection for Salmon Freshness During Cold Storage"

_foods, 2020, doi:10.3390/foods9111579_

Round 1
Reviewer 1 Report
The authors took into account my previous comments. You can also see the new quality of the presented work. As this work has been re-submitted for review, I believe that it meets the conditions for recommending it for further stages of evaluation.
Reviewer 2 Report
I think that now the article has been made suitable for publication
This manuscript is a resubmission of an earlier submission. The following is a list of the peer review reports and author responses from that submission.
Round 1
Reviewer 1 Report
The very idea of using an electronic nose to gauge food quality is not new. It is also difficult to find a scientific novelty. The authors of the manuscript did not critically analyze whether similar applications were made by other scientists. There is no information on the parameters of the electronic nose. It seems to me that the work needs to undergo a thorough reconstruction to further evaluate it. Below are my other comments:
page 2, line 52-57, I believe there is a need to add a paragraph on food quality assessment with the electronic nose, its use, advantages and disadvantages compared to other food quality measurement methods.
page 2, line 75-77, No information has been given on whether this type of fish quality detection system has been used in other studies? The use of the electronic nose for food analysis is very high, so where's the new science? where is the advantage over other applications?
page 3, line 87-88, not specified what gas sensors were used? secondly, what was the concentration level of the analyzed gases and what were the gases? So what was the gas mixture above the salmon sample. How were the sensors calibrated? in relation to what.
page 6, line 157-159, formula 2 is common knowledge, it need not be introduced or described.
page 7, line 180-182, the description of the PCA method is well known, it is the simplest method of data analysis. There is no need to show in Fig. 7 how the PCA method works.
Author Response
Review report 1
Comments:
The very idea of using an electronic nose to gauge food quality is not new. It is also difficult to find a scientific novelty. The authors of the manuscript did not critically analyze whether similar applications were made by other scientists. There is no information on the parameters of the electronic nose. It seems to me that the work needs to undergo a thorough reconstruction to further evaluate it. Below are my other comments:
page 2, line 52-57, I believe there is a need to add a paragraph on food quality assessment with the electronic nose, its use, advantages and disadvantages compared to other food quality measurement methods.
Response: Thank you for the comments. We have add a table on food quality assessment with the electronic nose. The advantages and limitations of e-noses application on food assessment is listed. Please see page 2, line 45-53.
Comments:
page 2, line 75-77, No information has been given on whether this type of fish quality detection system has been used in other studies?
Response: Thank you for the comments. The electronic nose detection system was developed by our lab. In the literature, metal oxide semiconductor gas sensors have been used for food quality detecting. We have listed the application object in Table 1.
The use of the electronic nose for food analysis is very high, so where's the new science? where is the advantage over other applications?
Response: Thank you for the comments. I agree with you. The use of the electronic nose for food analysis is very high. However, the use of electronic nose is from lab made of this work. The electronic nose detection system of this work has the advantages of small- sized and low cost. The sensor array consists of MQ136, MQ 137, MQ 138, TGS2612, TGS822 and TGS2600 sensors. It's very easy and feasible to detect the samples.
Comments:
page 3, line 87-88, not specified what gas sensors were used? secondly, what was the concentration level of the analyzed gases and what were the gases? So what was the gas mixture above the salmon sample. How were the sensors calibrated? in relation to what
Response: Thank you for the comments. We have added the specific information of gas sensors, mainly including O2, CO2, H2S sensor. The O2 concentration level ranges from 0 – 25%, CO2 concentration level ranges from 0 – 5000ppm, H2S concentration level ranges from 0 – 50ppm. Gas mixture above the salmon sample mainly includes aldehydes, alcohols and hydrocarbons, which are detected by electronic nose.
The sensors are tested and calibrated with standard gas at laboratory temperature of 25℃and relative humidity of 30%, O2, CO2 and H2S sensor measured the output voltage under the condition of different concentration ratios. Please see page 3, line 80-88.
Comments:
page 6, line 157-159, formula 2 is common knowledge, it need not be introduced or described.
Response: Thank you for the comments. We have deleted it.
Comments:
page 7, line 180-182, the description of the PCA method is well known, it is the simplest method of data analysis. There is no need to show in Fig. 7 how the PCA method works.
Response: Thank you for the comments. We have reduced the text.
Reviewer 2 Report
The article is well written, and, although the analytical procedures are well conducted, the technical practice is too predominant respect to a scientific experimental framework.
In my opinion Authors should be more synthetic in the exposition and should better clarify the main goal to be tested (freshness prediction by machine learning), the corresponding experiment design and the significant experimental procedures that conducts to the machine learning model.
Less text space should be dedicated to technical details or to standard procedures as PCA or electronic nose and so on. The main goal (prediction of freshness) should clearly specified and developed withdrawing details about accessory procedures.
Article length should be shortened and the number of tables or figures should be lowered, limiting them to the ones strictly pertinent to the main study goal. Description of intermediate steps to reach the main goal should be minimized. Machine learning procedures should be described in a simpler way making clear their advantages on the standard procedures.It should be useful to compare the machine learning model to predict freshness to a statndard model (for example multivariate score outcome regression).
So I recommend to the Authors to rewrite the article following the above indications in order to highlight the demonstration of the efficacy of the proposed technology.
Author Response
Review Report 2
Comments:
The article is well written, and, although the analytical procedures are well conducted, the technical practice is too predominant respect to a scientific experimental framework. In my opinion Authors should be more synthetic in the exposition and should better clarify the main goal to be tested (freshness prediction by machine learning), the corresponding experiment design and the significant experimental procedures that conducts to the machine learning model. Less text space should be dedicated to technical details or to standard procedures as PCA or electronic nose and so on. The main goal (prediction of freshness) should clearly specified and developed withdrawing details about accessory procedures. Article length should be shortened and the number of tables or figures should be lowered, limiting them to the ones strictly pertinent to the main study goal. Description of intermediate steps to reach the main goal should be minimized. Machine learning procedures should be described in a simpler way making clear their advantages on the standard procedures.It should be useful to compare the machine learning model to predict freshness to a statndard model (for example multivariate score outcome regression). So I recommend to the Authors to rewrite the article following the above indications in order to highlight the demonstration of the efficacy of the proposed technology.
Response: Thank you for the comments. We have reduced the text of standard procedures as PCA or electronic nose, and we have added more detailed steps and description of machine learning model to predict freshness.
With kind regards
ZHANG Xiaoshuan, On behalf of co-authors
Round 2
Reviewer 1 Report
I agree with most of the answers given to me by the authors of the manuscript.
The revised manuscript keeps getting better, but I still see room for improvement. Two more comments below:
- Table 1, although the advantages and disadvantages of the electronic nose in food analysis are described, I believe that the authors did not take into account two great works that summarize the use of the electronic nose in food analysis:
a.) Electronic noses: Powerful tools in meat quality assessment, Meat Science, 131 (2017), pp. 119-131
b.) Applications of electronic noses and tongues in food analysis, International Journal of Food Science and Technology, 39 (6) (2004), pp. 587-604.
- I believe that in chapter 2.1.2. you need to add information on the detection limits of sensors: MQ136, MQ 137, MQ 138, TGS2612, TGS822 and TGS2600
Author Response
Comments:
- Table 1, although the advantages and disadvantages of the electronic nose in food analysis are described, I believe that the authors did not take into account two great works that summarize the use of the electronic nose in food analysis:
a.) Electronic noses: Powerful tools in meat quality assessment, Meat Science, 131 (2017), pp. 119-131
b.) Applications of electronic noses and tongues in food analysis, International Journal of Food Science and Technology, 39 (6) (2004), pp. 587-604.
Response: Thank you for the comments. We have added summary and references based on these two great work. Please see page 2.
Comments:
I believe that in chapter 2.1.2. you need to add information on the detection limits of sensors: MQ136, MQ 137, MQ 138, TGS2612, TGS822 and TGS2600.
Response: Thank you for the comments. We have added the related information of sensors. Please see page 4, line 94-102.
Reviewer 2 Report
In my opinion no real efforts were made to make the article more effective - in practice the information was communicated in the same way as the previous version. The length of the article has not really been reduced (in my opinion the length should be halved, also removing irrelevant tables and figures that have been kept instead).
Author Response
Comments: In my opinion no real efforts were made to make the article more effective - in practice the information was communicated in the same way as the previous version. The length of the article has not really been reduced (in my opinion the length should be halved, also removing irrelevant tables and figures that have been kept instead).
Response: Thank you for the comments. We have reduced the text and removed irrelevant tables and figures. Please see the revised manuscript.